# Oxidized Alginate Dopamine Conjugate: A Study to Gain Insight into Cell/Particle Interactions

**DOI:** 10.3390/jfb13040201

**Published:** 2022-10-25

**Authors:** Adriana Trapani, Filomena Corbo, Erika Stefàno, Loredana Capobianco, Antonella Muscella, Santo Marsigliante, Antonio Cricenti, Marco Luce, David Becerril, Stefano Bellucci

**Affiliations:** 1Department of Pharmacy-Drug Sciences, University of Bari “Aldo Moro”, I-70125 Bari, Italy; 2Dipartimento Scienze e Tecnologie Biologiche e Ambientali, University of Salento, I-73100 Lecce, Italy; 3ISM-CNR, Via del Fosso del Cavaliere 100, I-00133 Rome, Italy; 4Laboratori Nazionali di Frascati, Istituto Nazionale di Fisica Nucleare, Via Enrico Fermi 54, Frascati, I-00044 Rome, Italy

**Keywords:** conjugates, dopamine, SH-SY5Y cell viability, SNOM microscopy, antioxidant activity

## Abstract

**Background**: We had previously synthetized a macromolecular prodrug consisting of oxidized Alginate and dopamine (AlgOx-Da) for a potential application in Parkinson disease (PD). **Methods**: In the present work, we aimed at gaining an insight into the interactions occurring between AlgOx-Da and SH-SY5Y neuronal cell lines in view of further studies oriented towards PD treatment. With the scope of ascertaining changes in the external and internal structure of the cells, multiple methodologies were adopted. Firstly, fluorescently labeled AlgOx-Da conjugate was synthetized in the presence of fluorescein 5(6)-isothiocyanate (FITC), providing FITC-AlgOx-Da, which did not alter SH-SY5Y cell viability according to the sulforhodamine B test. Furthermore, the uptake of FITC-AlgOx-Da by the SH-SY5Y cells was studied using scanning near-field optical microscopy and assessments of cell morphology over time were carried out using atomic force microscopy. **Results**: Notably, the AFM methodology confirmed that no relevant damage occurred to the neuronal cells. Regarding the effects of DA on the intracellular reactive oxygen species (ROS) production, AlgOx-Da reduced them in comparison to free DA, while AlgOx did almost not influence ROS production. **Conclusions**: these findings seem promising for designing in vivo studies aiming at administering Oxidized Alginate Dopamine Conjugate for PD treatment.

## 1. Introduction

The deficiency of the neurotransmitter DA in the *Substantia Nigra* is well known to be the key-factor in the neurological disorder of PD. Several investigations have been addressed to supply DA to the brain compartment by circumventing the Blood Brain Barrier (BBB), which inhibits its direct transport because of the polar features of the chemical structure of the neurotransmitter. In recent years, nanoparticulate dosage forms, capable to physically entrap DA, have been studied, starting not only from polymers [1,2,3,4] but also from lipid matrices [5,6,7]. The resulting nanostructured systems have paid attention to the protection exerted towards the early autoxidation of DA, which takes place at a physiological pH value and, additionally, they could ensure a slow release of the neurotransmitter. Overall, irrespectively of the type of nanocarrier selected, nanoparticle capabilities, to prevent enzymatic attacks and protect DA from chemical degradation, are both beneficial to keep the structure of the neurotransmitter intact. Among the routes of administration investigated for non-invasive DA supply to the brain, nanoparticles made of lipids, the biopolymer chitosan (CS, or CS derivatives) were seen to be favorable to the internalization occurring via the nasal route, providing the so-called nose-to-brain pathway [6,8,9]. However, the risk of DA loss from a nanostructured carrier during manipulation or fabrication cannot be excluded because no covalent linkage strongly connects the neurotransmitter to the excipient(s) of the nanodevice. Moving from this context, the strategy of designing high molecular weight prodrugs (also called conjugates) is currently a promising alternative to the pharmaceutical nanocarriers administering active substances, which also includes DA. To design suitable conjugates, some requirements should be fulfilled, among which, the selected polymer should be neither inherently toxic nor immunogenic or unstable under different pH conditions. Furthermore, if the polymer is non-degradable it must be characterized by a low molecular weight with a rather narrow polydispersity, which are both favorable to renal elimination and, hence, its progressive accumulation in the body is prevented [10,11]. Additionally, to achieve satisfactory drug concentrations from the high-molecular-weight prodrug, many reactive side groups on the main chain are essential. Moreover, in terms of designing pharmaceutical dosage forms for innovative drug delivery, the relevance of the macromolecular prodrugs is to be intended not only for their use as such, but also because they can be nanostructured if amphiphilic domains exist, and they can guarantee self-assembly in a colloidal particle size [12]. For instance, in cancer treatments, the approach of macromolecular prodrugs has been extensively reviewed [13,14,15,16] although in the literature, several examples of low molecular weight prodrugs have been conceived for PD treatment [17,18,19], but few examples of conjugates of DA have been reported [11,20,21,22,23,24]. Recently, the capability of the polymeric DA conjugate to disaggregate fibrils of a-synuclein into oligomers has been paid attention to, being a strategy of high impact [22]. In this context, we have recently described a novel conjugate based on aldehyde oxidized Alginate linked to DA via a Schiff base Alginate (AlgOx-Da, [25]). It was intended for a nose-to-brain delivery approach applied to PD and in vitro preliminary characterizations have already been described. Herein, with the aim to focus on the preliminary biological effects of the AlgOx-Da conjugate, investigations on cell/conjugate interactions were performed and discussed, including cell viability, uptake experiments, and Radical Oxygen Species (ROS) formation. In particular, it is well known that the morphological changes induced by engineered materials in cellular samples are crucial to evaluate their impact on human health. Indeed, the interaction study centered on polymeric prodrugs and cellular molecules requires cutting-edge instrumentation and dedicated protocols. For this scope, atomic force microscopy (AFM) and Scanning Near-Field Optical Microscopy (SNOM) are both relevant imaging techniques that possess nanometer resolution and three-dimensional observation, as well as the peculiarities of high resolution and a direct relationship with 3D cellular morphology and chemical evaluation of interacting groups. Therefore, AFM and SNOM visualization techniques have been herein adopted to achieve images, from which the interaction mode between the DA polymeric prodrug and the neuronal SH-SY5Y cell monolayer can be elucidated.

## 2. Materials and Methods

### 2.1. Materials

Dopamine hydrochloride (DA), dimethylsulphoxide, sodium dodecyl sulphate, and fluorescein 5(6)-isothiocyanate (FITC) were purchased from Sigma-Aldrich (Milan, Italy). Dialysis tubes with a MWCO 3500 Da and 1200–14,000 Da were provided from Spectra Labs (Rome, Italy). Oxidized Alginate (*M*_w_: 30,000–60,000 g/mol; polydispersity ≅ 2) was prepared as described elsewhere [23]. Throughout this work, double distilled water was used. All other chemicals used were of a reagent grade.

### 2.2. Preparation of Fluorescent AlgOx-Da

On the basis of two previous protocols, which were slightly edited [11,25], 30 mg of AlgOx-Da were dissolved in 3 mL of double distilled water, after which, the pH value was adjusted to 5 using HCl 0.1N. Then, 1 mL of an ethanolic solution of FITC (40 mg/mL) was added and stirred for 24 h under dark conditions at r.t. The mixture was then dialyzed in water (3 days) and then lyophilized for 72 h (Lio Pascal 5P, Milan, Italy).

### 2.3. Quantitative Analysis of DA and FITC and Determination of FITC-AlgOx-Da Degree Substitution (DS)

HPLC analysis was adopted for DA quantification [26]. The composition of the mobile phase consisted of 0.02 M potassium phosphate buffer, pH 2.8: CH_3_OH 70:30 (*v*/*v*), and the elution of the column took place in the isocratic mode at a flow rate of 0.7 mL/min. Under such chromatographic conditions, the retention time of DA was found to be equal to 5.5 min, whereas retention time of AlgOx-Da was equal to 4.9 min.

To determine the labeling efficiency (namely, the percent weight of FITC to weight of the FITC-AlgOx-Da conjugate) of the FITC-AlgOx-Da conjugate, an aliquot of 2 mg was then initially dissolved in 0.1 M phosphate buffer, pH 8.0. Standard solutions of 1 to 140 ng/mL of FITC in phosphate buffer, pH 8.0, were obtained from the dilution of a buffered stock solution of 100 μg/mL FITC in methanol (excitation and emission wavelengths of 488 and 525 nm, respectively, slits, 2.5 cm) occurred when using fluorometer (Perkin Elmer, Milan, Italy) calibration.

Furthermore, to determine the substitution degree (DS), the FITC-AlgOx-Da conjugate underwent acid hydrolysis [25]. Two mg of the FITC-AlgOx-Da conjugate were weighed and dissolved in 2 mL of HCl 1 N (pH 1), stirred and protected from light, at r.t. for 3 h. Then, the resulting mixture was centrifuged (16,000× *g*, for 45 min, Eppendorf 5415D, Hamburg, Germany) and, for the obtained supernatant, the HPLC analyses allowed for the quantification of the DA levels. The DS was calculated as mg DA/mg FITC-AlgOx-Da imine conjugate.

### 2.4. Cell Culture

The SH-SY5Y cells were grown in a DMEM medium (Sigma, St. Louis, MO, USA) complete with streptomycin (100 mg/mL), penicillin (100 U/mL), 10% heat-inactivated fetal bovine serum (FBS) and glutamine 2 mM, in a humidified incubator containing 5% CO_2_ in air at 37 °C. In subsequent experiments, the cells were always used at passage three.

### 2.5. Cytotoxicity Assay via Sulforhodamine B

By using the sulforhodamine B (SRB) assay [27,28] we studied the inhibition of the SH-SY5Y cell proliferation in different experimental conditions. Cells at 70–80% confluency were treated with 0.25% trypsin, washed and then resuspended in growth medium. We added one hundred µL of cell suspension (10^5^ cells/mL) to each well of a 96-well plate, and after overnight incubation, the cells were incubated for 24 h with various concentrations of FITC-AlgOx-Da (9–450 µg/mL) and DA (1–50 μg/mL). After these incubations, the SRB assay was performed as previously described [25].

### 2.6. Fluorescence Microscopy

The SH-SY5Y cells were seeded on cover glass and placed at the bottom of 35 mm Petri dishes at a density of 3 × 10^5^ cells per dish. Cells at 70–80% confluence were washed with PBS and then treated with FITC-AlgOx-Da (25 μg/mL) for 30 min at room temperature. Afterwards, the cells were washed twice with PBS and fixed in 4% paraformaldehyde for 10 min. Subsequently, they were washed again twice with PBS, dehydrated in graded alcohols (50, 75 and 100% ethanol), cleared in xylene, and mounted with coverslips in Eukitt^®^, a quick-hardening mounting medium (Fluka^®^Analytical, Milan, Italy).

### 2.7. ROS Generation

The ROS generation was detected by colorimetric nitro blue-tetrazolium (NBT, Thermo Fisher Scientific, Milan, Italy) assay which consisted of the reduction of yellow water-soluble tetrazolium chloride by superoxide to an insoluble violet di-formazan [29,30].

The SH-SY5Y were treated with DA, AlgOx-Da at different time points. After treatment, the culture medium was replaced with an unconditioned medium and NBT (1 mg/mL) was added; treated cells were incubated at 37 °C for 1 h, then carefully washed and lysed in a 90% dimethylsulphoxide solution containing 0.01 N NaOH and 0.1% sodium dodecyl sulphate. Absorbance of formazan was measured at 620 nm using a SpectroStar spectrophotometer (BMG Labtech, Ortenberg, Germany) and data were expressed as a % of control untreated cells.

### 2.8. Atomic Force Microscopy (AFM)-SNOM Visualization

The features of the home-designed AFM instrument were described elsewhere [31]. Room temperature AFM images in air were collected in the weak repulsive regime of the contact mode, with a force of less than 1 nN from the zero cantilever deflection. The samples were air dried, to allow for a detailed AFM analysis spanning over several days, which was the minimum time required in order to collect a statistically meaningful set of data for each sample. Triangular shaped Bruker’s Sharp Silicon Nitride Microlever were used with a spring constant of 0.01 N/m and a radius of curvature of 10 nm. This force is very low and does not induce deformations or damages to biological samples [32,33,34]. Constant force images were collected with a typical scan rate of 2–4 s/row (800–1600 points/row).

In the acquired data, the background was subtracted and the same cell was observed twice in order to exclude possible damage from tip interaction and experimental artifacts.

Topographic and optical (fluorescence and absorption) experiments were performed simultaneously by an AFM/SNOM apparatus built on top of an inverted optical microscope [35]. Several sets of experiments were performed and different areas of the sample were imaged with an optical absorption at 405 nm and a fluorescence emission above 500 nm. The inverted microscope was used to check features in the dispersions of FITC-AlgOx-Da at different FITC concentrations in water and, once pin-point areas were detected, then high resolution SNOM images were required. The topographical and optical were performed on dried FITC-AlgOx-Da samples and deposited onto a glass substrate.

### 2.9. Statistical Analysis

Statistical analyses were performed with GraphPad Prism software Version 4, GraphPad Software Inc. (San Diego, CA, USA) using the ANOVA associated with Tukey’s multiple comparisons test. A *p*-value less than 0.05 was considered to achieve statistical significance.

## 3. Results

### 3.1. Determination of FITC-AlgOx-Da Substitution Degree (DS)

The fluorescent FITC-AlgOx-Da conjugate was produced as described in Section 2.2 with a yield of 51% (±6). The amount of DA covalently bound to the FITC-AlgOx-Da imine conjugate resulted from hydrolysis under strong acidic conditions, as described in Section 2.3 and it was found to be equal to 11 (±0.9) μg of DA/mg of FITC-AlgOx-Da imine conjugate. By a fluorometric analysis, the DS of FITC in the fluorescent conjugate FITC-AlgOx-Da was found to be equal to 36 (±6 × 10^−6^) μg of FITC/mg of FITC-AlgOx-Da.

### 3.2. Effects of DA and FITC-AlgOx-Da on SH-SY5Y Viability

In this study, the cytotoxic effect of DA on the SH-SY5Y cells was confirmed and the inhibitory effect of DA on the SH-SY5Y cell viability was significantly decreased when the FITC-AlgOx-Da conjugate was administered (Figure 1). In fact, when the concentration of FITC-AlgOx-Da was between 260 and 1300 µg/mL, the cell viability remained quite high (between 90% and 80%), while with concentrations of FITC-AlgOx-Da below 260 µg/mL, cell viability did not change significantly when compared to the control. As a matter of fact, at 260 µg/Ml FITC-AlgOx-Da the concentration of free neurotransmitter corresponded to 10 µg/Ml and, at this concentration, the DA administered as such to the cells caused 60% of cell viability (namely, 40% of the cells died, Figure 1).

### 3.3. FITC-AlgOx-Da Detection by Fluorescence Microscopy

The SH-SY5Y neuroblastoma cells were incubated with. 25 μg/mL of FITC-AlgOx-Da for 30 min at room temperature and then were observed by fluorescence microscopy. The fluorescent conjugate was revealed at the plasma membrane level of the cells (Figure 2).

### 3.4. ROS Generation

The ROS generation has been demonstrated to be a common feature occurring in DA-treated cells. In this study, we examined the effect of AlgOx-Da on the ROS generation in an SH-SY5Y cell model line. While exposure of the SH-SY5Y cells to 10 μg/mL of pure DA led to a 2.2-fold increase when compared with the control cells, the AlgOx-Da conjugate caused a much smaller increase (1.4-fold) in the intracellular ROS production (Figure 3). Notably, incubation with 0.3 mg/mL of AlgOx alone caused no significant change to the ROS production in the SH-SY5Y cells (Figure 3).

### 3.5. Atomic Force Microscopy (AFM) Observations

AFM measurements (Figure 4a,b) were taken on many isolated cells and the observations allowed to obtain topography images that were representative of the cells within a mean line-profile A-A’, evidencing the general shape and height of the cells. Such cell shapes and their height mean values did not change significantly, along with uptake of FITC-derivatives within the reported values for the exposure time.

In this context, all dimensions, height, and roughness were also analyzed and subsequently displayed in Figure 4c,d [36,37].

Figure 4d shows that varying cell heights, the local cell roughness has an important increase, ranging from 19 nm for control cells, to values between 21 and 34 nm for 10 to 120 min exposure times, indicating an appreciable variation in the membrane level of FITC-AlgOx-Da uptake.

Additionally, Figure 5 includes photos from a 120-min FITC exposure and displays a fluorescent SNOM image, together with the corresponding topographic image (Figure 5a) and transmission image (Figure 5b), of the FITC-AlgOx-Da sample at different exposure times, ranging from 10 to 120 min. In the fluorescence SNOM image (Figure 5c), green areas correspond to stronger emission from the fluorophore, while in the transmission images (Figure 5b), dark areas correspond to a stronger absorption from the material. The topographic image shows cells with a width of several microns and a height of 0.55 μm, which were in good agreement with the AFM higher resolution images. Similar topographic images were acquired in several areas and were representative of the cell structure.

A similar fluorescence behavior was observed with all samples at different exposure times, even though we observed several clusters of precipitates on top of cells. A statistical analysis of the fluorescence intensity for FITC-AlgOx-Da (0–120 min) samples, according to SNOM observations, is summarized in Figure 5d.

The fluorophore emission area observed in the fluorescent SNOM images for the analyzed cells in the different samples was continuously increasing as a function of the FITC exposure times, with a sharp increase in the case of the 120 min exposure sample. This indicated a much higher accumulation of FITC inside the cells in the case of 120 min exposure.

## 4. Discussion

The bypass of the BBB still remains a challenge, with the P-glycoprotein efflux pump [38,39] remaining an obstacle, which limits drug access to the CNS and, therefore, some modifications to the natural polymeric backbone are required to try to overcome such limitations. For instance, the chemically modified alginate backbone leads to oxidized alginate, which, due to its favorable features, has been often introduced into hydrogels, microspheres, 3D-printed/composite scaffolds, membranes, and electrospinning and coating materials [40]; additionally, oxidized alginate-based materials can be easily functionalized and can deliver drugs or growth factors in view of tissue regeneration [38].

In this context, we had characterized an AlgOx-Da imino conjugate in our previous work [25], which proposed a nose-to brain-delivery method, to address the neurological disorder of PD. To the best of our knowledge, this is a unique case of a DA conjugate via Schiff base to AlgOx and, hence, we were highly motivated to gain an insight into the interactions occurring between such polymeric prodrugs and the selected neuronal cell model line represented by SH-SY5Y. Firstly, for visualization purposes, we had synthetized the fluorescent FITC-AlgOx-Da imino-conjugate, which resulted in a DS in the neurotransmitter DA, similar to one of the unlabeled AlgOx-Da (11 μg of DA/mg of FITC-AlgOx-Da vs 12 μg of DA/mg of AlgOx-Da, respectively [25]). Interestingly, when preliminary reactions of FITC labeling were performed on the backbone of AlgOx-Da, by setting the dye concentration at 5 mg/mL alcohol, around 1 μg FITC/mg FITC-AlgOx-Da conjugate was obtained, which was not suitable for fluorescence microscopy. Therefore, the FITC concentration for condensation was increased to 40 mg FITC/alcohol, owing to the fact that the ligand FITC should be available at higher initial concentrations in order to be introduced to the backbone of the already substituted AlgOx-Da. Finally, upon these conditions, we obtained the DS of 11 μg FITC/mg FITC-AlgOx-Da, which was used for the cell/conjugate interaction studies.

Firstly, FITC-AlgOx-Da was evaluated for cytotoxicity towards the SH-SY5Y cell line as shown in Figure 1, evidencing that up to 260 μg/mL of FITC-AlgOx-Da (corresponding to a DA concentration equal to 10 μg/mL) no significative effect on the SH-SY5Y cell viability was shown. Precisely, at a FITC-AlgOx-Da concentration equal to 260 μg/mL, 90% of the initial cells survived, whereas, for the sake of comparison, the previously studied FITC-AlgOx resulted in a safer biomaterial because, upon incubation for 24 h with a concentration of 450 g/mL, it did not affect the SH-SY5Y cell viability [25]. We believe that the DA covalently bound to FITC-AlgOx-Da determines the reduction in the SH-SY5Y cell viability, but, on the other hand, it should be pointed out, that when the FITC-AlgOx-Da was set to 260 μg/mL (i.e., DA concentration equal to 10 μg/mL) the cytotoxicity reached 10%, which was significantly lower than cell mortality observed with the free DA at 10 μg/mL, which determined a cell mortality of 40% (Figure 1). Indeed, from these results it can be deduced that the high cytocompatibility of the natural sodium alginate was retained in the AlgOx backbone and also when it was bound to a chemical compound. Moreover, it is in good agreement with a recent study, where a novel interpenetrating polymer network of alginate/gelatin hydrogels formed the scaffold for the encapsulation of SHSY-5Y cells, with full safety of the cells addressed to the in vitro three-dimensional cultures and organ bioprinting [41].

Once assessed, the cell viability of FITC-AlgOx-Da can be usefully employed to ascertain the localization of the SHSY-5Y cell monolayer thanks to fluorescence microscopy (Figure 2, [42]), leading to the conclusion that the labeled conjugate was placed at the plasma membrane level rather than close to the cell nuclei.

Furthermore, the interactions AlgOx-Da conjugate/ SH-SY5Y cells were investigated in terms of the ROS production [43] because, ROS levels, once increased, can also contribute to the determination in the death of DAergic neurons [44]. When the cells were exposed to AlgOx for 24 h, no statistically significant difference in terms of the ROS production was detected, whereas AlgOx-Da significantly reduced ROS production when compared with free DA, especially at the longest time points of incubation (Figure 3). Overall, both AlgOx and AlgOx-Da evidenced to some extent an antioxidant role, which is an important feature in terms of application to PD, where mitochondrial dysfunction plays a crucial role to DA neurons’ degeneration together with DA deficiency [44]. Hence, further investigations are planned to gain an insight into the visualization of the interactions of AlgOx-Da conjugate/SH-SY5Y cells and whether some damage can be ascribed to the AlgOx-Da conjugate.

In our previous study [25], we had already investigated the FITC-AlgOx samples deposited onto a glass substrate by using SNOM optical nanospectroscopy. The fluorescent AlgOx samples were deeply investigated by looking at its structure in different parts of the sample. Herein, the SNOM characterization of the FITC-AlgOx-Da fluorescent effect evidenced a homogeneous distribution within the conjugate at several concentrations (Figure 5).

In detail, the fluorescence spectroscopic SNOM image of the FITC-AlgOx-Da showed emissions distributed in areas that corresponded to the position of the cells in the topographic image and the highest optical emission was always localized in correspondence with the cell position (Figure 5, [45]).

All the acquired data allowed for the assessment of a strong correspondence between topographic structures of cells and fluorescence maxima, suggesting the incorporation of the fluorophore FITC inside the cells. It is worth observing that there were some clusters in the mid-lower part of the topographical image that corresponded to a strong absorption at 405 nm (dark areas). However, no fluorophore was embedded within these clusters, as clearly seen in the fluorescence image where no emission was observed in the corresponding areas.

Additionally, the fluorescence SNOM images in Figure 5c evidenced that a half of the total cells possessed a clear fluorescent signal and, more precisely, a non-uniform fluorescence distribution was detected throughout the cells. It is worth pointing out that no area, with a significant fluorescence emission, was present in the corresponding topographic image where no cell was observed.

Together with SNOM visualization, AFM measurements (Figure 4) were performed on all samples to evaluate the quality of the sample preparation and possible changes in the cells induced by the uptake of the FITC conjugate. Preliminary AFM images were collected in the tapping mode, but the lateral resolution was worse, so inducing us to abandon such an approach. Interestingly, our results on several tens of cells showed that they were well fixed, quite stable and showed the following changes within the limits of the experimental error. In particular, the mean height (Figure 4c) ranges from 0.58 μm for control cells to values of 0.5 and 0.6 μm for 10 to 120 min of exposure time, respectively, so suggesting that the uptake of the FITC-derivative does not change the general structure of the SH-SY5Y cells at the cytoskeleton level [46,47].

## 5. Conclusions

AlgOx-Da imine conjugate, labelled and unlabelled, was subjected to in vitro cell studies to assess the safety of such conjugates in view of a potential application for PD treatment. In detail, the conjugate did not alter the SH-SY5Y cell viability plus the conjugate did not increase dramatically the ROS. In combination with such biological investigations, AFM and SNOM spectroscopies clarified that the neuronal SH-SY5Y cells were not modified in their structure upon incubation with AlgOx-Da. Overall, high levels of AlgOx-Da biocompatibility were herein demonstrated, which discloses the perspectives to study its performance in vivo.

## Figures and Tables

**Figure 1 jfb-13-00201-f001:**
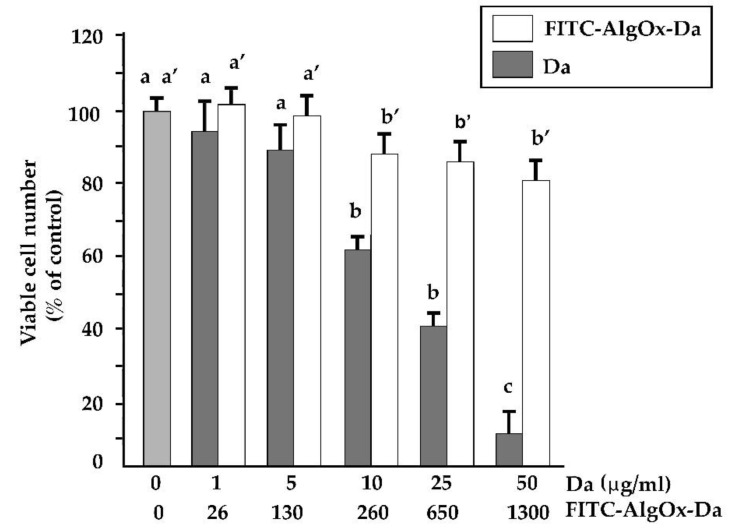
The sensitivity of SH-SY5Y cells to DA and FITC-AlgOx-Da. Cells were treated with increasing concentrations of DA (1 to 50 μg/ML) and FITC-AlgOx-Da for 24 h. The data are means ± S.D. of three different experiments, with eight replicates for each experiment, and are presented as percentages of control. Values of the histograms with shared letters are not significantly different according to Bonferroni/Dunn’s post hoc tests.

**Figure 2 jfb-13-00201-f002:**
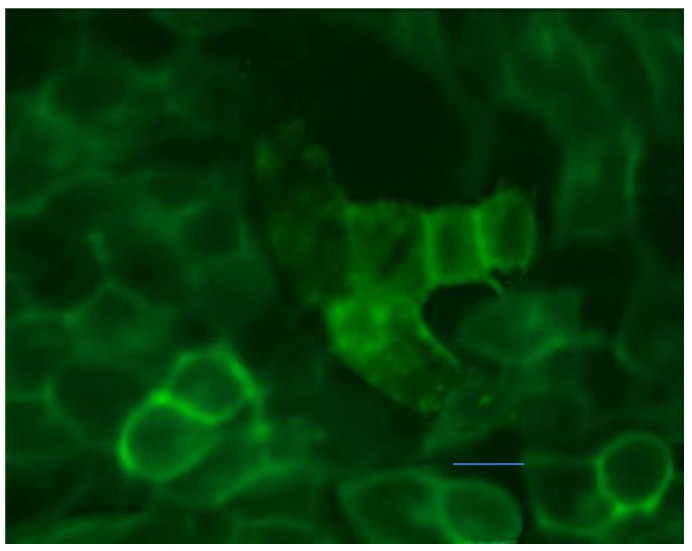
Localization of FITC-AlgOx-Da by fluorescence microscopy in SH-SY5Y cells treated with 25 μg/mL of FITC-AlgOx-Da for 30 min. Representative fluorescence microscopy images of four independent experiments (*n* = 4). Scale bar: 5 μm.

**Figure 3 jfb-13-00201-f003:**
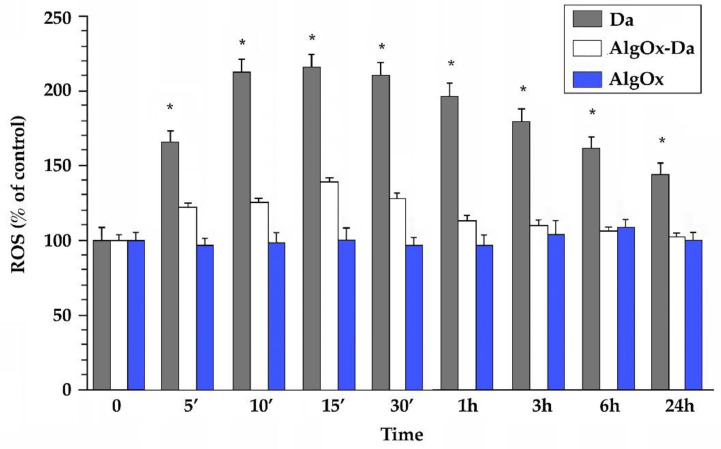
ROS generation according to the colorimetric NBT assay. SH-SY5Y cells were treated with DA (10 μg/mL), AlgOx-Da (DA 10 μg/mL and AlgOx 0.3 mg/mL) and AlgOx (0.3 mg/mL), for different time points (from 5 min to 24 h). The data are means ± S.D. obtained from three independent experiments performed with eight replicates in each and are presented as % of control. Asterisks indicate values that were significantly different between DA and AlgOx-Da concentration (Student’s *t*-test, * *p* < 0.05).

**Figure 4 jfb-13-00201-f004:**
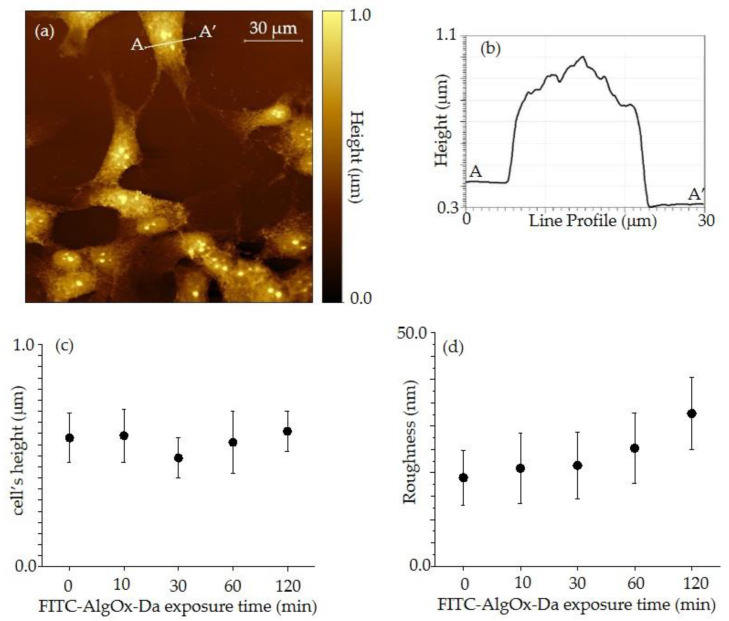
Contact mode AFM image of control SH-SY5Y cells (**a**) and a corresponding line-profile on the white line A-A’ along an isolated cell (**b**); dependence of cell height from FITC exposure time (**c**) and dependence of cell roughness from FITC exposure time (**d**).

**Figure 5 jfb-13-00201-f005:**
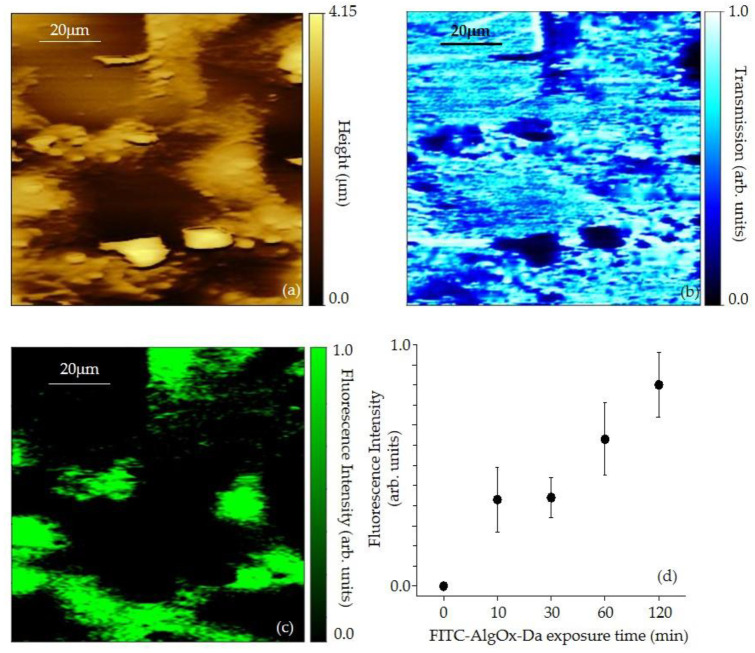
SNOM topography (**a**), transmission (**b**), and fluorescence images of FITC-AlgOx-Da (1 mg/mL), (**c**) deposited on the glass by micropipette. Statistical analysis of FITC-AlgOx-Da (0 to 120 min FITC exposure time) fluorescence emission from cells according to SNOM observations (**d**).

## Data Availability

Not applicable.

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
