# Peer review of "Oxidized Alginate Dopamine Conjugate: A Study to Gain Insight into Cell/Particle Interactions"

_jfb, 2022, doi:10.3390/jfb13040201_

Round 1

Reviewer 1 Report

The study is interesting and fits the scope of JFB. It should be accepted once the following issues are solved.

1. Line 66-67, several other studies (Acta Biomaterialia 83 (2019) 31; Polymers 2020, 12(5), 1132) related to dopamine conjugates should be included.

2. Line 100, the molecular weight and polydispersity of oxidized alginate should be added.

3. Why don't the authors make a quantitative analysis of DA and FITC substitution degree by NMR?

4. Technical issues. Line 114, 'CH3OH' to 'CH3OH'. Please check all.

Author Response

The study is interesting and fits the scope of JFB. It should be accepted once the following issues are solved.

We are grateful to the Reviewer for his/her opinion on our work

Q1. Line 66-67, several other studies (Acta Biomaterialia 83 (2019) 31; Polymers 2020, 12(5), 1132) related to dopamine conjugates should be included.

R1. Both references recommended by the Reviewer were inserted in the revised version of the manuscript

Q2. Line 100, the molecular weight and polydispersity of oxidized alginate should be added.

R2. Molecular wight details of oxidized alginate were added in the revised version

Q3. Why don't the authors make a quantitative analysis of DA and FITC substitution degree by NMR?

R3. Our work was conceived as a study focusing on cell/conjugate interactions and, hence, a deep chemical characterization of the fluorescent conjugate was considered out of the scope of the research herein performed

Q4. Technical issues. Line 114, 'CH3OH' to 'CH3OH'. Please check all.

R4. The correction was performed in the revised manuscript

Reviewer 2 Report

It is a well-written study related to oxidized Alginate dopamine conjugate. The results are new and the topic is interesting. I recommend it for publication after the issues below are well addressed.

1. In Figure 2, the bright field image must be added.

2. For the AFM scanning of cells, non-contact mode (taping mode) would be better. 

3. Does the FITC modification change the physical-chemical properties of the alginate?

Author Response

It is a well-written study related to oxidized Alginate dopamine conjugate. The results are new and the topic is interesting. I recommend it for publication after the issues below are well addressed.

We acknowledge the Reviewer for appeciating our manuscript

Q1. In Figure 2, the bright field image must be added.

R1. When bright field image was recorded, indeed it did not look well and, therefore, we did not show it in the manuscript

 Q2. For the AFM scanning of cells, non-contact mode (taping mode) would be better. 

R2. As reported in the revised version of Discussion Section, similar images have been collected in tapping mode, but the lateral resolution was worse.

Q3. Does the FITC modification change the physical-chemical properties of the alginate?

R3. In the present work we aimed at elucidating the interactions occurring between the fluorescent conjugate and SH-SY5Y cell model line and a deep investigation focusing on the physical-chemical properties was omitted

Reviewer 3 Report

AlgOX-DA imine conjugate, both labeled and unlabeled, was put through in vitro cell experiments in this study to evaluate its safety in light of its possible use as a therapy for Parkinson's disease. The article contains sufficient characterization to support its concept, but there are some minor problems that need to be fixed before moving on.

1.      Beginning with either a capital letter or lowercase letter for terms oxidize and dopamine can produce consistency.

2.      Uptake and imaging microscopy are not appropriate keywords. Please provide alternative keywords.

3.      Typographical and grammatical mistakes are on lines 43, 83, 222, 234, 255, 303, 343, and more. Please carefully revise.

4.      The writers neglected to remove the journal template's last paragraph (lines 86-94).

5.      The preparation of fluorescent algOX-DA was described in the materials and methods section as using 40 mg/mL FITC, and Figure 2 shows the localization of FITC-AlgOX-DA by fluorescence microscopy in SH-SY5Y cells treated with 25 g/mL of FITC-AlgOX-DA. Could the writers please explain their actions?

6.      There is no Figure 3 provided.

7.      "In this context, all dimensions, height, and roughness have also been analyzed and subsequently displayed in Figure 5 [34,35]," the authors write on line 267 of Figure 5. "Additionally, Figure 5 includes photos from a 120-minute FITC exposure and displays a fluorescent SNOM image (Figure 5c)", as is stated on line 273 of the document. There appears to be a mistake in the first sentence's description of the figure.

8.      The authors' description of the cytotoxicity of FITC-AlgOX-DA towards the SH-SY5Y cell line is difficult to grasp. Can the authors sum up and evaluate their results in light of previous research?

9.      Overall, both AlgOX and AlgOX-DA demonstrated modest antioxidant activity, according to scientists. Have the authors performed further quantitative or qualitative studies in addition to those described in this study to show the advantages of DA functionalization if both materials have comparable qualities in terms of cell response?

10.  The text font size and style in the conclusion are different, and it can be written as a single paragraph.

Author Response

AlgOX-DA imine conjugate, both labeled and unlabeled, was put through in vitro cell experiments in this study to evaluate its safety in light of its possible use as a therapy for Parkinson's disease. The article contains sufficient characterization to support its concept, but there are some minor problems that need to be fixed before moving on.

 Q1.      Beginning with either a capital letter or lowercase letter for terms oxidize and dopamine can produce consistency.

R1. The new nomenclature for the conjugate of Dopamine was adopted in the revised manuscript

Q2.      Uptake and imaging microscopy are not appropriate keywords. Please provide alternative keywords.

R2. “SH-SY5Y cell viability” and “SNOM microscopy” replaced the previous keywords in agreement with the Reviewer’s suggestion

Q3.      Typographical and grammatical mistakes are on lines 43, 83, 222, 234, 255, 303, 343, and more. Please carefully revise.

R3. The Reviewer’s corrections were performed in the revised version

Q4.      The writers neglected to remove the journal template's last paragraph (lines 86-94).

R4. We apologize for this mistake. The Journal template's last paragraph was deleted in the revised version.

 Q5.      The preparation of fluorescent algOX-DA was described in the materials and methods section as using 40 mg/mL FITC, and Figure 2 shows the localization of FITC-AlgOX-DA by fluorescence microscopy in SH-SY5Y cells treated with 25 g/mL of FITC-AlgOX-DA. Could the writers please explain their actions?

R5. The concentration of FITC was set at 40 mg/mL FITC to force the fluorescent ligand covalent linkage onto the backbone of DA-conjugate. For fluorescence micriscopy purposes, SH-SY5Y cells treated with 25 mg/mL of FITC-AlgOX-DA

 Q6.      There is no Figure 3 provided.

R6. The Figure 3 was now provided

 Q7.      "In this context, all dimensions, height, and roughness have also been analyzed and subsequently displayed in Figure 5 [34,35]," the authors write on line 267 of Figure 5. "Additionally, Figure 5 includes photos from a 120-minute FITC exposure and displays a fluorescent SNOM image (Figure 5c)", as is stated on line 273 of the document. There appears to be a mistake in the first sentence's description of the figure.

R7. We aknowledge the Reviewer for this issue. In the revised version, in Paragraph 3.5, according to the Reviewer’s suggestions, corrections were done.

 Q8.      The authors' description of the cytotoxicity of FITC-AlgOX-DA towards the SH-SY5Y cell line is difficult to grasp. Can the authors sum up and evaluate their results in light of previous research?

R8. We acknowledge the Reviwer for this observation and, in the revised version, Paragraph 3.2 was edited accordingly

Q9.      Overall, both AlgOX and AlgOX-DA demonstrated modest antioxidant activity, according to scientists. Have the authors performed further quantitative or qualitative studies in addition to those described in this study to show the advantages of DA functionalization if both materials have comparable qualities in terms of cell response?

R9. No further study has been done than that shown in the manuscript. However, we would like to focus the Reviewer's attention on the important fact of the result shown in Figure 3, namely that DA administered alone to the cells doubled the quantity of ROS generated by the cells themselves, while AlgOx-Da did not increase such compounds.

Q10.  The text font size and style in the conclusion are different, and it can be written as a single paragraph.

R10. The Reviewer’s requirement was matched

Reviewer 4 Report

The authors have investigated the safety of previously synthesized AlgOX-DA.

Despite the high quality of this work, in general, I have some questions:

- Could the authors explain the physiological significance of in-air AFM (and SNOM? If yes, it should be described in the methods) of cells?

-  Error bars on the AFM and SNOM quantification graphs should be added.

- Absence of Fig 3. 

Author Response

The authors have investigated the safety of previously synthesized AlgOX-DA.

Despite the high quality of this work, in general, I have some questions:

Q1. Could the authors explain the physiological significance of in-air AFM (and SNOM? If yes, it should be described in the methods) of cells?

R1. As reported in the revised version of Paragraph 2.8., the samples were air dried, to allow a detailed AFM analysis spanning over several days, which was the minimum time required in order to collect a statistically meaningful set of data for each sample

Q2. -  Error bars on the AFM and SNOM quantification graphs should be added.

R2. Error bars have been added in the appropriate panels of Figure 4 and 5 to accomplish the Reviewer’s recommendations

Q3. Absence of Fig 3. 

R3. We are afraid for this issue and in the resubmission process Figure 3 was now provided as a separate JPEG file

Round 2

Reviewer 4 Report

The authors have significantly increased the quality of this article.

Author Response

We are grateful to the Reviewer for his/her assessment